# Developing serum proteomics based prediction models of disease progression in ADPKD

Hande Ö. Aydogan Balaban[1,2,6], Sita Arjune [1,2,3,6], Franziska Grundmann [1], Jan-Wilm Lackmann [4], Thomas Rauen[5], Philipp Antczak [1,2,4] & Roman-Ulrich Müller [1,2,3,4]

Autosomal Dominant Polycystic Kidney Disease is the most common genetic cause of kidney failure. Outcome prediction is essential to guide therapeutic decisions. However, currently available models are of limited accuracy. We aimed to examine the potential of serum proteomics for improved risk stratification. Here we show that 29 proteins are significantly associated with yearly kidney function decline. Functional enrichment on these 29 proteins reveals GO:BP terms related to immune response, lipoproteins and metabolic processes. A comparison to an Immunoglobulin A nephropathy cohort provides information regarding the eGFR-dependency and disease specificity of these proteins. The final outcome prediction model (adjusted $R^2$ 0.31) contains six proteins, namely Endothelial Plasminogen Activator Inhibitor (SERPINF1), Glutathione Peroxidase 3 (GPX3), Afamin (AFM), FERM Domain Containing Kindlin-3 (FERMT3), Complement Factor H Related 1 (CFHR1), and Retinoic Acid Receptor Responder 2 (RARRES2), the predictive value of which is independent from the clinical and imaging parameters currently used in clinical care. The validation of these models in different cohorts indicates the accuracy of the models. It will now be important to move towards targeted validation in a prospective study.

ADPKD is an inherited disorder characterized by the development of multiple cysts in both kidneys, leading to renal enlargement and a gradual decline in renal function[1]. It is the most common genetic cause of kidney failure and affects millions of people worldwide[2–4].

Prediction of future outcome and specifically the question when kidney replacement therapy (KRT) will be required is of central importance to individualized patient counseling and therapeutic decision-making. The only targeted treatment available, the vasopressin-2

receptor antagonist Tolvaptan, is only approved for patients with rapid disease progression[5,6]. Potentially lifelong exposure of patients to side effects such as polyuria and the associated expense to healthcare systems are only warranted in cases with an expected benefit of this therapy in prolonging the time to kidney failure. Besides, stratification of patients for clinical trials regarding upcoming therapeutic opportunities relies on models predicting future decline in kidney function measured as decline in estimated glomerular filtration rate (eGFR decline).

[1]Department II of Internal Medicine, University of Cologne, Faculty of Medicine and University Hospital Cologne, Kerpener Str. 62, 50937 Cologne, Germany. [2]Center for Molecular Medicine Cologne (CMMC), Robert-Koch-Str. 21, 50931 Cologne, Germany. [3]Center for Rare Diseases Cologne, Faculty of Medicine and University Hospital Cologne, University of Cologne, Kerpener Str. 62, 50937 Cologne, Germany. [4]Cologne Excellence Cluster on Cellular Stress Responses in Aging-Associated Diseases (CECAD), University of Cologne and University Hospital Cologne, Joseph-Stelzmann-Str. 26, 50931 Cologne, Germany. [5]Department of Nephrology and Clinical Immunology, RWTH Aachen University Hospital, Pauwelsstraße 30, 52074 Aachen, Germany. [6]These authors contributed equally: Hande Ö. Aydogan Balaban, Sita Arjune. ✉e-mail: philipp.antczak@uk-koeln.de; roman-ulrich.mueller@uk-koeln.de

Currently available models are either based on kidney volumetry, as integrated in the Mayo Imaging Classification (MIC), or use clinical complications and genetic information (e.g. PROPKD score) for patient stratification[7,8]. However, all available models provide limited accuracy only, a crucial shortcoming considering the central importance of this information. Furthermore, genotype and MRI-based volumetry may not always be easily available at the point of presentation of a patient. Consequently, the recent KDIGO guideline on ADPKD has clearly identified the need for better and easily accessible biomarkers of ADPKD progression, e.g. blood-bourne, biomarkers adding to predictive accuracy as an important goal for the field[9]. This study uses a dedicated semi-automated mass spectrometry pipeline for the unbiased quantification of serum proteins as a source of biomarkers using samples from two large cohorts to improve the prediction of eGFR decline in ADPKD.

## Results

For objective measurement of serum proteins as potential biomarkers, a specialized semi-automated mass spectrometry process was developed and employed to measure all serum samples in this study. For the Screening Cohort (SC), serum samples from 264 patients with ADPKD

### Table 1 | Clinical baseline characteristics

| | Screening Cohort (SC) | Internal/Temporal Cohort (ITC) | External Cohort (EC) |
|---|---|---|---|
| **Patients, *n*** | 214 | 408 | 173 |
| **Sex (Female %)** | 56.5 | 57.4 | 57.8 |
| **Age (years), median (IQR)** | 46.1 (16.9) | 48.2 (17.7) | 46.6 (17.5) |
| **eGFR (ml/min/m²), median (IQR)** | 67.3 (44.5) | 65.0 (51.7) | 66.0 (46.6) |
| **TKV (ml), median (IQR)** | 1315.5 (1300.0) | 1479.0 (1399.8) | 1415.0 (1335.0) |
| **Mayo classification, *n*** | 212 | 392 | 169 |
| 1A | 2 | 14 | 8 |
| 1B | 62 | 118 | 33 |
| 1C | 75 | 140 | 68 |
| 1D | 58 | 87 | 41 |
| 1E | 15 | 33 | 19 |
| **CKD stage, *n*** | 214 | 408 | 173 |
| 1 eGFR ≥ 90 ml/min/1.73m² | 56 | 108 | 47 |
| 2 eGFR 60-89 ml/min/1.73m² | 71 | 116 | 55 |
| 3 eGFR 30–59 ml/min/1.73m² | 80 | 123 | 54 |
| 4 eGFR 15-29 ml/min/1.73m² | 7 | 51 | 17 |
| 5 eGFR <15 ml/min/1.73m² | 0 | 10 | 0 |
| **Hypertension, *n*** | 152 | 161 | 171 |
| No | 22 | 22 | 41 |
| Yes | 130 | 139 | 130 |
| **Genotype, *n*** | 115 | 175 | 162 |
| PKD1, non-truncating | 19 | 37 | 55 |
| PKD1, truncating | 64 | 79 | 75 |
| PKD2 | 32 | 59 | 32 |

*CKD* Chronic Kidney Disease, *eGFR* Estimated Glomerular Filtration Rate, *SD* Standard Deviation, *TKV* Total Kidney Volume, *IQR* Interquartile Range. For ITC, *n* provided is the number of samples instead of patients since it employs several time points per patient.

derived from the German AD(H)PKD registry were characterized using an in-house proteomics workflow resulting in 214 patients after quality control, outlier removal and filtering for slope (Fig. S1). A second sub-cohort was generated, including baseline samples of patients not contained in the SC and follow-up samples of overlapping patients (Internal/Temporal Cohort [ITC]: 623 samples from 465 patients in total, with 408 remaining samples representing 305 patients after quality control, outlier removal and filtering for slope, Fig. S1). For external validation, we similarly characterized 221 EDTA-plasma samples from 221 patients of the Dutch observational multicenter DIPAK study (External Cohort, EC) and quality control, outlier removal and filtering for slope resulted in 173 samples from 173 patients (Fig. S1). Median time of follow-up used for eGFR slope calculation across all three cohorts were 6.8, 6.4, and 6.0 for SC, ITC, and EC respectively (Table S1). Clinical baseline characteristics of these cohorts are summarized in Table 1 and Table S2.

### General characteristics of the serum proteome of SC

Comparing the mass distribution of all human proteins (UniprotKB) to the serum proteins detected in the serum proteome of the SC shows a distinct protein profile (Fig. S2A). Principal component analysis (PCA) indicates that, although sex does not significantly influence the first two components, eGFR appears to contribute more to the variation in the second component, as evidenced by the color gradient (Fig. S2B). The heatmap reveals distinct clustering patterns based on protein abundance intensity, with additional annotation for eGFR, Mayo imaging classification, age, and sex, indicating that eGFR and Mayo classification align with key changes in protein abundance across the dataset (Fig. S2C). To further examine the identified proteins, enrichment analysis was conducted, revealing enriched terms to be associated with immunological response, hemostasis, endocytosis, lipid metabolism, and cell adhesion and migration (Supplementary Data S1 and S2, Fig. S3).

### Identification of proteins associated with annual eGFR decline

The linear modeling technique, LIMMA, identified 27 proteins (LIMMA set) that exhibited a statistically significant relation with eGFR slopes (FDR adjusted *p*-value < 0.05, Fig. 1). Weighted LASSO (wLASSO) was employed to identify a selection of proteins (LASSO set, *n* = 6), emphasizing those with the strongest correlation to eGFR slope after controlling for patient variability. Four proteins were identified by both the LIMMA and wLASSO methods, namely Glutathione Peroxidase 3 (GPX3), Complement Factor H Related 1 (CFHR1), endothelial plasminogen activator inhibitor (SERPINF1), and Retinoic Acid Receptor Responder 2 (RARRES2).

The analysis of the integrated LIMMA and LASSO dataset (29 proteins) identified three enriched biological processes according to parent GO:BP terms: lipid digestion (GO:0044241), regulation of interleukin-1 production (GO:0032652), and negative regulation of multicellular organismal processes (GO:0051241). A comprehensive enumeration of enriched terms is available in Supplementary Data S3. The 29 proteins were subsequently arranged into a hierarchical heatmap, as illustrated in Fig. 1. All proteins depicted in the heatmap, except for Afamin (AFM) and FERM Domain Containing Kindlin-3 (FERMT3), exhibited a significant relation with eGFR slope as determined by LIMMA. The clustering analysis of proteins identified three unique clusters of protein abundance among the samples. Cluster 1 proteins consisted of enriched parent terms: embryonic organ morphogenesis (GO:0048562), nervous system process (GO:0050877), while Cluster 2 proteins consisted of only one enriched term: positive regulation of interleukin-1 production (GO:0032732). Cluster 3 contained diverse parent terms. Comprehensive and parent-enriched terms are included in Supplementary Data S4 and S5, respectively. Furthermore, patients were also categorized into

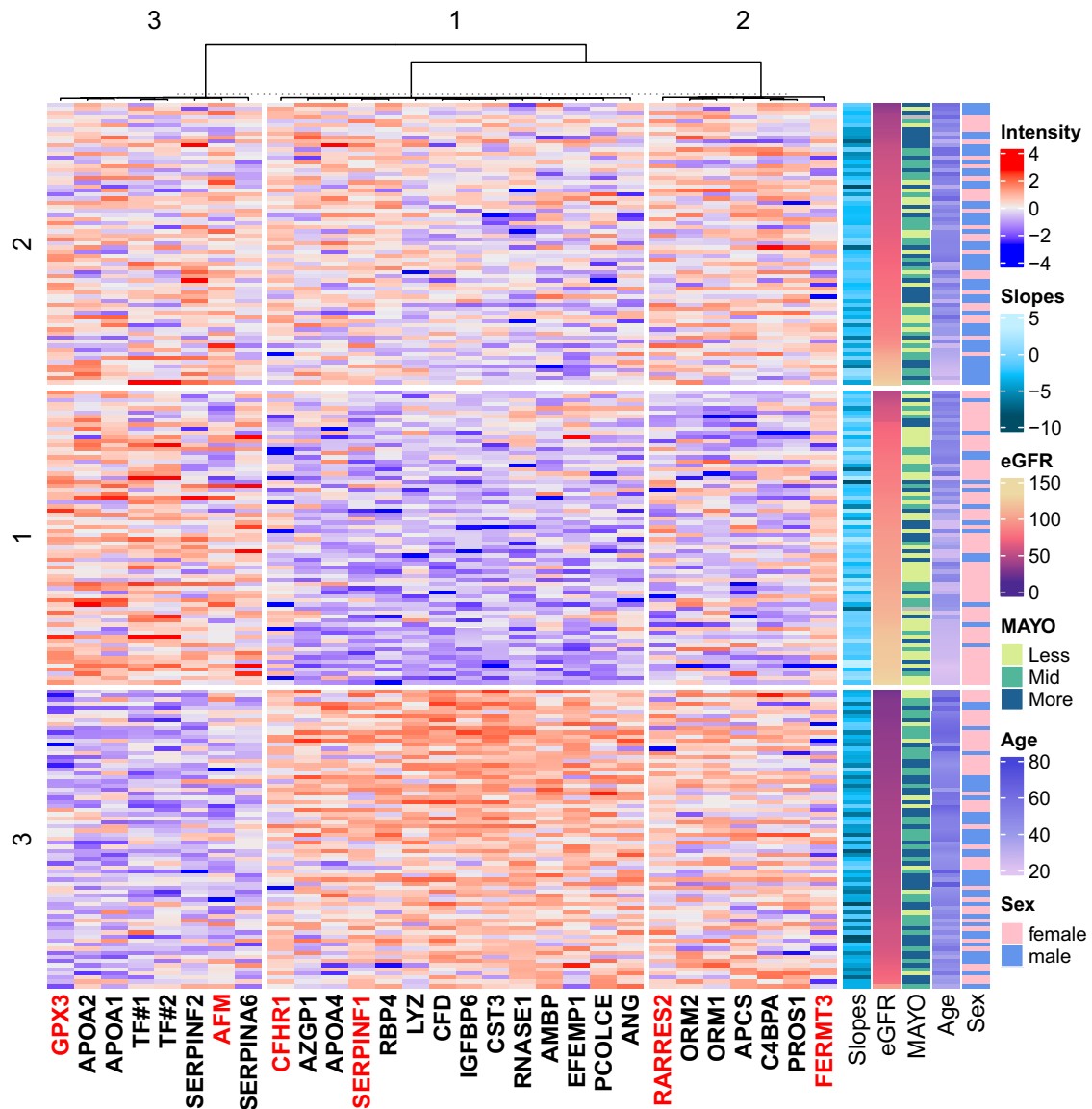

**Fig. 1 | Heatmap of the proteins associated with eGFR slopes.** Clustering was performed on both patients (rows) and proteins (columns). Each patient cluster was sorted according to eGFR (annotation on the right), separately. All listed proteins except AFM and FERMT3 are also selected by LIMMA (LIMMA set), while proteins marked red were selected by wLASSO (LASSO set). GPX3 = Glutathione Peroxidase 3, AFM = Afamin, FERMT3 = FERM Domain Containing Kindlin-3, CFHR1 = Complement Factor H Related 1, SERPINF1 = Endothelial Plasminogen Activator Inhibitor (Serpin F1), RARRES2 = Retinoic Acid Receptor Responder 2, MAYO MAYO Class, Intensity = Protein Abundance. Source data are provided as a Source Data file.

three distinct clusters. The characteristics of the identified patient clusters and the significant differences among groups are detailed in Table S3. To obtain information whether the 29 proteins depend on eGFR themselves, we correlated the eGFR at the time point of biosampling with the levels of these proteins (Table S4). This analysis showed—as expected for proteins associated with disease severity based on eGFR slope—a significant correlation for all proteins apart from ORM2. Importantly, only 19 of 28 proteins showed a negative correlation with eGFR, in principle in line with decreased metabolism/excretion, while 9 proteins showed a positive correlation. In addition, to address the question whether these findings are specific to ADPKD or driven by eGFR decline in general, we added data from a cohort of patients with IgA nephropathy (IgAN, Fig. S4, Table S5). Here, only 9 proteins showed a significant correlation with eGFR (ANG, GPX3, IGFBP6, CFD, SERPINF2, LYZ, PROS1, CST3, AMBP). Interestingly, these 9 proteins all show the same direction of correlation with eGFR in both patients with ADPKD and IgAN. Importantly, CST3, i.e. cystatin, is among the proteins with a significant negative correlation with eGFR in both cohorts serving as a proof-of-principle.

**Developing models predictive of eGFR slope in screening cohort**
Since wLASSO was tailored to mitigate the influence of extreme eGFR slope values on the selection process, only the LASSO ($n_{protein} = 6$) set was used to build a predictive linear regression model for eGFR slope, namely the Proteome Model (Table S6). To assess the robustness of the model, cross-validation was used. Median explained variance ($R^2$) was 0.29 (Fig. S5) with an average Root Mean Square Error (RMSE) of 2.08. These metrics were in concordance with the original Proteome Model (Table S6, RMSE: 2.03).

All six proteins exhibited independent significant associations ($p$-value < 0.05) with eGFR slope, with Endothelial Plasminogen Activator Inhibitor (SERPINF1), FERM Domain Containing Kindlin-3 (FERMT3),

**Table 2 | Outcome prediction models in the Screening Cohort**

| Predictors | Proteome Model | | | Clinical Model | | | Combined Model | | |
|---|---|---|---|---|---|---|---|---|---|
| | β | CI | p | β | CI | p | β | CI | p |
| (Intercept) | −2.74 | −3.01 – −2.47 | **<0.001** | −6.96 | −10.00 – −3.92 | **<0.001** | −3.58 | −6.73 – −0.42 | **0.027** |
| SERPINF1 | −0.63 | −0.96 – −0.30 | **<0.001** | | | | −0.37 | −0.74 – 0.01 | 0.055 |
| GPX3 | 0.71 | 0.42 – 1.00 | **<0.001** | | | | 0.58 | 0.26–0.90 | **<0.001** |
| AFM | 0.40 | 0.12–0.67 | **0.005** | | | | 0.27 | −0.02–0.57 | 0.069 |
| FERMT3 | −0.49 | −0.76 – −0.22 | **<0.001** | | | | −0.48 | −0.75 – −0.21 | **0.001** |
| CFHR1 | −0.34 | −0.61 – −0.06 | **0.016** | | | | −0.29 | −0.57 – −0.02 | **0.036** |
| RARRES2 | −0.33 | −0.63 – −0.03 | **0.030** | | | | −0.33 | −0.62 – −0.03 | **0.029** |
| Age | | | | 0.04 | 0.00–0.08 | **0.049** | 0.01 | −0.03–0.05 | 0.651 |
| Sex [Male] | | | | −0.52 | −1.11 – 0.08 | 0.089 | −0.47 | −1.05–0.11 | 0.110 |
| eGFR | | | | 0.04 | 0.03 – 0.06 | **<0.001** | 0.01 | −0.01 – 0.03 | 0.186 |
| MAYO [1 C] | | | | −0.48 | −1.22 – 0.26 | 0.205 | −0.20 | −0.90 – 0.50 | 0.577 |
| MAYO [1D-1E] | | | | −0.94 | −1.76 – −0.12 | **0.026** | −0.77 | −1.53 – 0.00 | 0.050 |
| Observations | 212 | | | 212 | | | 212 | | |
| R² / R² adjusted | 0.334 / 0.314 | | | 0.251 / 0.233 | | | 0.375 / 0.340 | | |

β = Estimates, CI = Confidence interval, p = p-value, SERPINF1 = Endothelial Plasminogen Activator Inhibitor (Serpin F1), GPX3 = Glutathione Peroxidase 3, AFM = Afamin, FERMT3 = FERM Domain Containing Kindlin-3, CFHR1 = Complement Factor H Related 1, RARRES2 = Retinoic Acid Receptor Responder 2, Age in years, eGFR in ml/min/1.73 m². Two-sided t-test without adjustment for multiple comparisons was performed. p-value < 0.05 is indicated in bold.

Complement Factor H Related 1 (CFHR1), and Retinoic Acid Receptor Responder 2 (RARRES2) showing a negative and Glutathione Peroxidase 3 (GPX3), and Afamin (AFM) a positive association. Interestingly, for RARRES2, which showed 22.2% missing values, the mere question of whether the protein was detected at all or not showed a significant negative correlation with eGFR slope.

**The proteome model surpasses clinical models and offers enhanced value when integrated with clinical and imaging data**
The Clinical Model incorporating age, sex, eGFR, and MIC (Table 2) was developed for comparison with existing current clinical standard of care. The Clinical Model explained 23.3% of the variance in the eGFR slope. Age and eGFR exhibited a strong positive correlation with the eGFR slope, a negative correlation was observed for male sex and elevated Mayo classes (1D and 1E). The integration of proteomic and clinical data improved the explanatory power (adjusted R² 0.340). Substantial associations (p-value < 0.05) with the eGFR slope were established for GPX3, FERMT3, CFHR1, and RARRES2, while marginal significance (p-value < 0.1) was observed for SERPINF1, AFM, and elevated Mayo classes; conversely, age, sex, and eGFR did not retain statistical significance.

The regression line for the Proteome Model was, as anticipated, slightly further from the line of identity than that of the Combined Model when scatter plots were employed to visually depict accuracies (Fig. 2A, B). We assessed the performance of our models at various stages by calculating the Root Mean Square Error (RMSE) for CKD Stages 1 and 2–4 (Fig. 2C). The Combined Model surpassed the Proteome Model with reduced RMSE values, albeit the disparity was marginal. Upon comparing the RMSEs for various CKD stages with our models, we observed that the RMSEs of the MIC Model were higher in both early and late-stage patients (Fig. 2C).

To complement the analyses for eGFR slope we compared the last eGFR value obtained for each patient to the value predicted by the different models (Fig. 2D). The Proteome and Combined Models demonstrated comparable abilities in predicting future eGFR values. The MIC Model showed a clear tendency to overestimate eGFR loss in the short term and underestimate eGFR loss after one year to a larger degree compared to the other models. The superior performance of the two proteome-containing models is also confirmed by the lower root mean squared error (RMSE, Fig. 2D).

Models integrating genotype were developed separately, considering the limited availability of genotype information and the resulting smaller cohorts. Adding genotype to the Clinical (i.e. containing clinical information and MIC) and Combined Model (i.e. containing clinical information, MIC and proteome) further augmented model performance (Table S7, adjusted R² 0.30 and 0.33, respectively).

**Validation of the proteome-based outcome prediction models**
To validate the models with data further analyses were performed on the ITC and the entirely external DIPAK cohort (EC). The DIPAK cohort utilized a distinct sample type (EDTA plasma), prompting inquiries over the potential predictive capacity of the serum proteome based models. To eliminate hemolysis as a potential cause of discrepancies between cohorts, we assessed the intensity of the hemoglobin beta subunit (HBB), finding no significant differences (Fig. S6).

The robustness of the Combined Model was assessed by evaluating its performance in the two validation cohorts through a comparison of predicted and observed slopes. The predictions of the Combined Model for ITC exhibited greater congruence with those of the SC than with the EC (Fig. 3, Fig. S7B). Additionally, the prediction accuracy of the Combined Model was assessed by comparing the slope predictions for the same patients at various time points. As shown in Fig. S7F, the majority of the predictions fall within one annual eGFR unit depicted by the dashed lines, indicating that the Combined Model yields comparable slope predictions for the same patients over time, irrespective of the chosen time point and cohort. Finally, the slope prediction accuracy of the Combined Model was confirmed by comparing its future eGFR predictions to those of the MIC Model in validation cohorts, as previously performed for the SC (Fig. 2D). The Combined Model still outperformed the MIC Model in ITC (Fig. S8A), consistent with SC. Conversely, the Combined Model exhibited performance comparable to the MIC model regarding EC (Fig. S8B).

Similar validation analyses were also performed on the Proteome Model and the Combined Genotype Model. For these models, we obtained better slope prediction in ITC compared to EC (Fig. S7A and S7C). As underlined by the regression lines, Combined Model's predictions were better than the Proteome Model, while Combined Genotype Model performs better than Combined Model. It is important to note that observation size decreased in the Combined Genotype

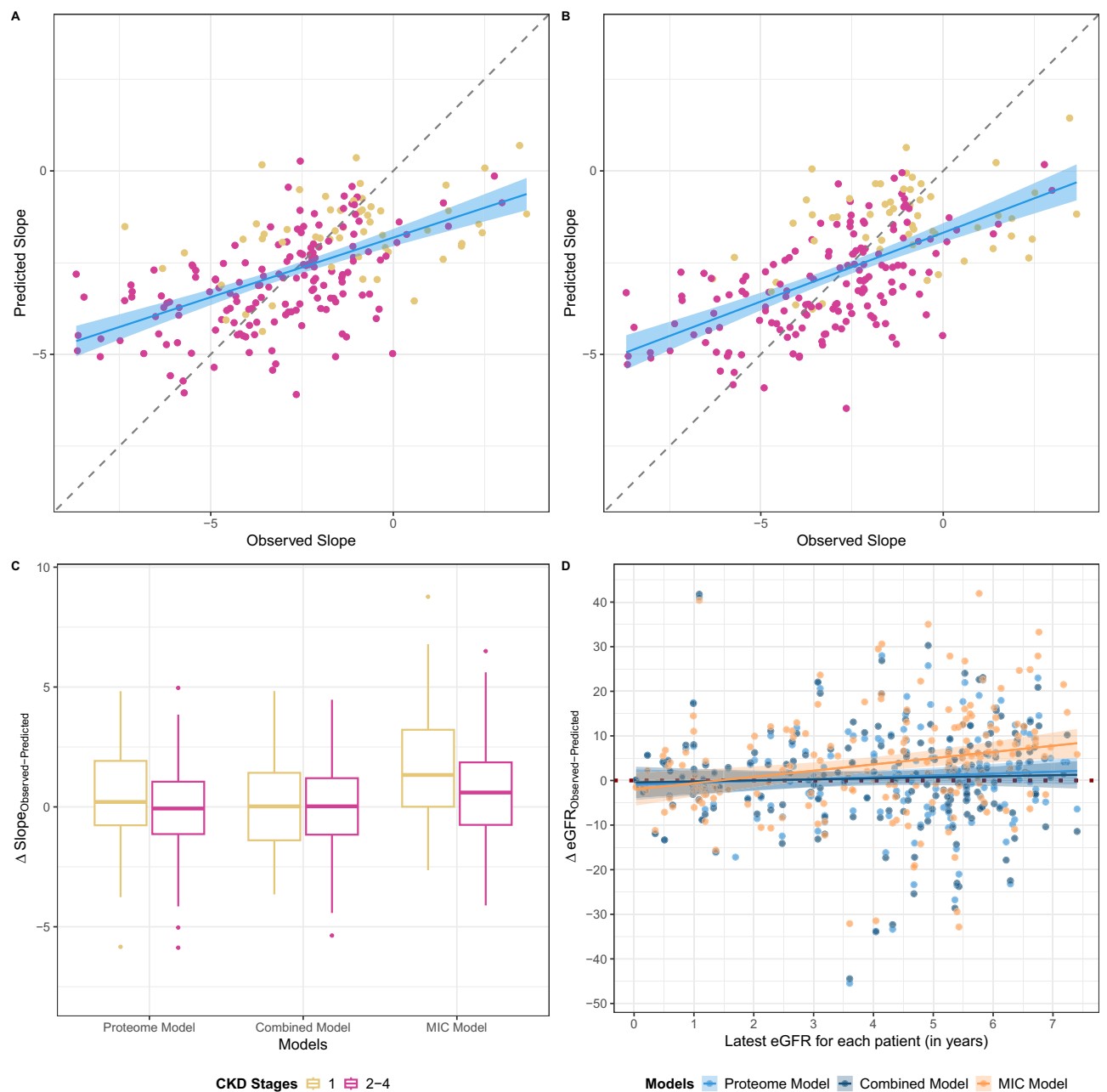

**Fig. 2 | Comparison of the slope predictions of the generated models in the Screening Cohort.** Observed slopes were plotted against predicted slopes according to (**A**) Proteome, and (**B**) Combined Model. The solid line is the fitted using a robust linear regression methodology and the shaded area shows bootstrapped 95% confidence interval. **C** The difference between observed and predicted slopes according to Proteome, Combined and MIC Models were box plotted and color-coded for different CKD stages. Patient numbers for each boxplot (from left to right) are 56, 158, 55, 157, 55 and 157. RMSEs for each boxplot (from left to right) are 2.23, 1.91, 2.11, 1.83, 3.09 and 2.11. The lower and upper bounds of the box are the 25th and 75th percentile of the data, the middle line indicates the median (50th percentile) and the whiskers are 1.5 * IQR. The outliers are plotted as dots.

**D** The plot shows the difference between observed and predicted eGFR values ($\Delta$eGFR = Observed - Predicted) for individual patients. Data points represent individual predictions, with models color-coded as follows: Proteome Model (blue, RMSE: 11.0, $n = 192$), Combined Model (dark blue, RMSE: 11.0, $n = 190$), and Mayo Imaging Classification (MIC) Model (orange, RMSE: 12.4, $n = 190$). The fitted regression lines and confidence intervals illustrate the error and variability in predictions for each model. A value of $\Delta$eGFR = 0 represents a perfect prediction, with deviations above or below indicating under- or overestimation of eGFR, respectively. The solid line is the fitted using a linear regression approach and the shaded area shows 95% confidence interval based on standard error. Source data are provided as a Source Data file.

Model (Table S8). Since two proteins from the panel of six contained in the original Proteome Model from the SC showed high percentages of missing values in the ITC and EC (RARRES2 and FERMT3, 74% and 66%, respectively), we developed an additional model excluding these two proteins (the Proteome4 Model). The elimination resulted in a slight reduction of explained variance while the significant independent connections were maintained (Table S9). Using this model, the

regression lines for SC and ITC move closer, even though—as expected based on the reduced $R^2$ - all regression lines move further away from the identity line (gray dashed line, Fig. S7D).

Assessment of the prediction accuracies of these three models (Proteome Model, Combined Genotype Model and Proteome4 Model) were performed by comparing the slope predictions for the same patients at various time points. As described for the Combined Model

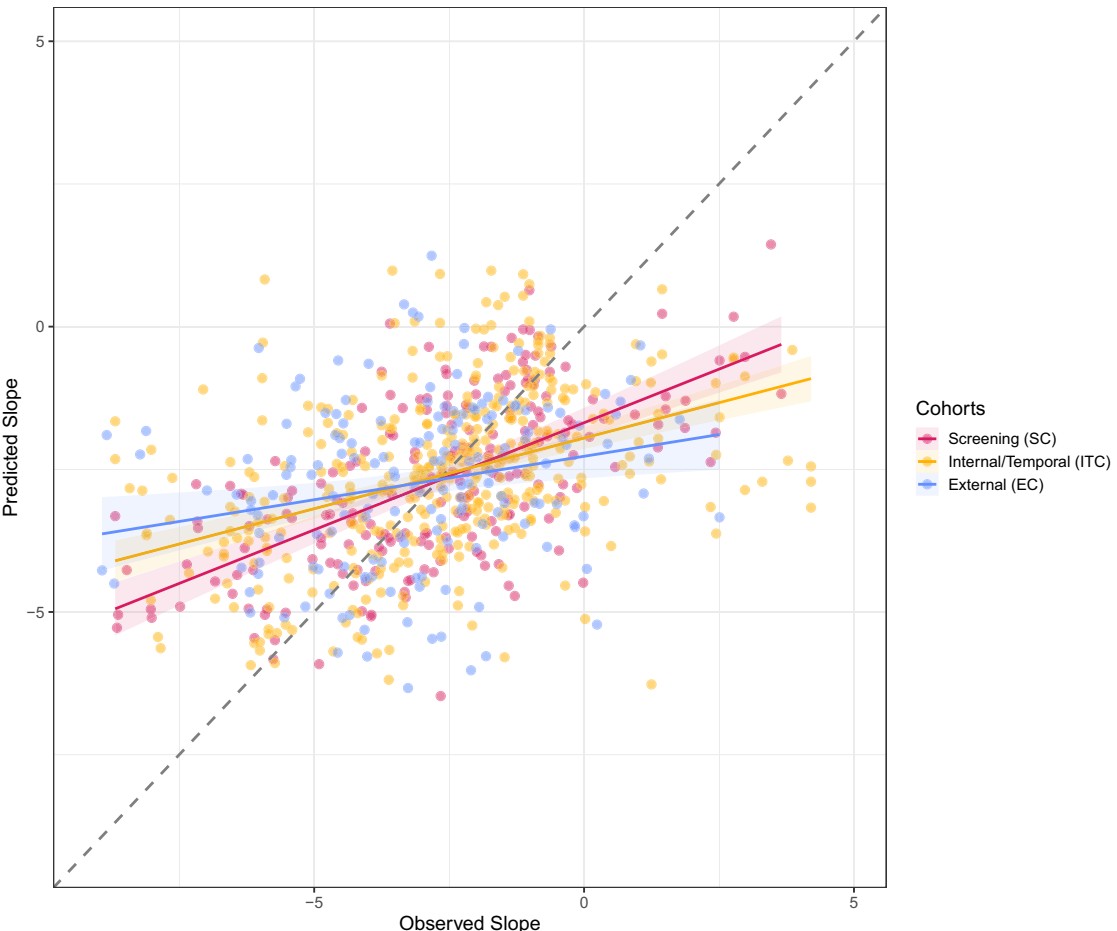

**Fig. 3 | Validation of the Combined Model.** The model was built on Screening Cohort ($n_{sample}$ = 212, red) and validated on Internal/Temporal ($n_{sample}$ = 392, yellow) and External ($n_{sample}$ = 169, blue) Cohorts. The solid line is the fitted using a robust linear regression methodology and the shaded area shows bootstrapped 95% confidence interval. Source data are provided as a Source Data file.

above, this approach again confirmed predictive capacity for all models over time (Fig. S7E–H) with the Proteome4 Model leading to the highest degree of similarity in model performance across cohorts. Finally, future eGFR predictions for the Proteome and Combined Model were compared to the MIC Model in the validation cohorts. In ITC, both models perform better than the MIC Model (Fig. S8A) and showed similar accuracy to the MIC Model in the EC. (Fig. S8B).

To examine the applicability of the proteome-based model to predict the clinically relevant cutoff for eGFR slope of > 3 ml/min/1.73 m²/year as defined by the KDIGO guideline[5,9], we generated ROC curves showing an area under the curve (AUC) of 0.82 and 0.69 for the Combined and the MIC Model, respectively (Fig. S9).

## Discussion

The recent KDIGO ADPKD guideline identifies urine and serum biomarkers as a key asset considering the costs and complexities of kidney imaging and genotyping in ADPKD. Consequently, this area is specifically listed as a future research topic highlighting the need for models integrating clinical data with such biomarkers[9]. Proteomics approaches are increasingly discussed to define biomarker panels including ADPKD[10–12]. However, while proteomics are a valid approach for broad screening of biological fluids, clinical implementation of the results has remained limited[13]. In line with this, and as for many other biomarker studies, most of these approaches never make it beyond the merely descriptive level. This is often due to limited cohort size and lack of comparison to existing clinical outcome prediction models to prove added value. Besides, missing validation cohorts is another

central limitation. The semi-automated MS pipeline developed and employed in this study allowed for the high-throughput measurement of large cohorts and increased comparability through detailed operating standards and the use of calibration samples between runs. This was combined with very well-characterized cohorts, enough outcome data and solid comparison to standard-of-care prediction models.

Using this approach, we identified six proteins (GPX3, SERPINF1, CFHR1, RARRES2, AFM, and FERMT3) to predict ADPKD disease progression. Interestingly, each of these proteins hold biological significance in the context of ADPKD pathogenesis. Glutathione peroxidase 3 (GPX3) plays a critical role in the oxidative stress response[14], a process that has been implicated in cyst growth and kidney damage in ADPKD[15]. SERPINF1 (also known as pigment epithelium-derived factor, PEDF) is known for its anti-angiogenic properties[16], and its diminished levels may reflect a dysregulation of vascular homeostasis within the cystic kidney. Besides, proteins of the SERPIN family have been found in cystic fluid[17]. CFHR1 (Complement Factor H-Related Protein 1) modulates complement activation[18], a finding which is of interest considering previous data suggesting that immune and inflammatory pathways are involved in disease progression[19]. RARRES2 (retinoic acid receptor responder protein 2, also known as chemerin) is a chemoattractant implicated in inflammation and adipogenesis[20,21], both of which may affect cyst development and fibrosis. AFM (Afamin) is involved in vitamin E transport[22], and may—as GPX3—be linked to oxidative stress in renal tissue. FERMT3 (kindlin-3) plays a vital role in integrin activation and cell adhesion[23,24], processes that are likely altered in the cellular

microenvironment of ADPKD. While the predictive capacity of these proteins for eGFR slope is independent from eGFR at the time point of biosampling, as shown by the multiple linear models, knowing which of them shows eGFR dependent changes in abundance would still be of interest when considering their potential pathophysiological impact. Interestingly, only one (GPX3) shows eGFR dependence in a separate cohort of patients with IgAN measured using the same pipeline, while the other 5 (AFM, CFHR1, SERPINF1, RARRES2, FERMT3) are not significantly correlated with eGFR. To gain a broader insight into existing data on the 6 biomarkers identified, we performed a thorough literature review on these proteins. While, not unexpectedly, changes in these proteins upon loss of kidney function have previously been implicated by other studies[25–33], directionality is not always identical and, in broader CKD cohorts, opposite to our findings for AFM[34]. Besides, to our knowledge, FERMT3 had not been reported in relation to CKD to date. Importantly, most of the available studies are small in cohort size and will require further validation. Furthermore, data on association with eGFR slope is only available for AFM and RARRES2[35], and, for the latter, is, in published work, oppositely related compared to our findings.

Interestingly, cystatin 3 (CST3), a recognized parameter for eGFR assessment in clinical use, was also identified as associated with eGFR slope in our analysis but was omitted by wLASSO due to multicollinearity with other proteins, including SERPINF1, CFHR1, and GPX3 (Table S10). Substituting CFHR1 or SERPINF1 with CST3 did not enhance the explained variance, while substituting GPX3 with CST3 deteriorated the fit (Table S11). While the incorporation of CST3 slightly enhanced the adjusted $R^2$ in the new model (Table S11, Model 4), its presence or absence did not significantly affect the overall performance of the model. This may also point towards the added value of the identified protein markers in eGFR estimation itself, a point which will need dedicated analyses in the future.

Our Proteome Model indicated that diminished levels of SERPINF1, FERMT3, CFHR1, and RARRES2, together with elevated levels of GPX3 and AFM, correlated with a reduced annual reduction in eGFR. The Proteome and Combined Models showed strong performance when the measured eGFR slope was within the anticipated range of −5 to 0 ml/min/1.73 m²/year for ADPKD. Nonetheless, their prediction accuracy diminished for slopes beyond this range, with the Combined Model exhibiting a slight advantage across all three cohorts. The better performance within the expected range of kidney function loss may speak for the fact that actual ADPKD-associated eGFR decline can well be predicted by serum proteome and existing models, while fluctuations − as likely for increases in kidney function - or additional causes of kidney damage, e.g. AKI, are less amenable to predictive models. The enhancement in the Combined Model's adjusted $R^2$ indicates a synergistic effect from integrating proteomic data with clinical factors, rendering it a more comprehensive predictor of annual eGFR reduction. However, incorporating clinical data into the Proteome Model diminished the significance of certain proteins, suggesting a common variation between clinical traits and proteins of reduced significance.

Importantly, the MIC Model is currently the clear gold-standard for outcome prediction in ADPKD as underlined by the recent KDIGO guideline[9]. Comparative analyses of our models against the MIC Model for prospective endpoint eGFR prediction indicated that both the Proteome and Combined Models surpassed the MIC Model in SC and ITC. Despite its nature as the best currently available predictor, the limited $R^2$ for the models containing clinical parameters and TKV (MIC) is indeed not unexpected. For instance, a recent study involving 618 patients showed that MIC could generally predict kidney failure and eGFR decline rates, though there was notable interindividual variability within each class[36] and previous publications reported comparable or even lower $R^2$-values for TKV-containing models[37,38] than the ones reported in our manuscript. In our study, all three models underestimated future eGFR; however, the extent of underestimation

differed. In the EC, the Proteome and Combined Models performed worse than in SC and ITC. This is likely to be explained by the use of EDTA-plasma instead of serum underlining that the protein panels described should be measured in serum in future trials. Nonetheless, interestingly, the proteome−despite this marked difference between cohorts−fmaintained predictive information and the resulting models performed comparably to the MIC Model.

The slope prediction of the Proteome Model for the same patients in SC and ITC, i.e. same patients, but different time points of sampling, was fairly high. Some of the remaining discrepancies may actually be explained by imputation of missing values. Removing the two proteins with the highest fraction of missing values in the ITC (RARRES2 and FERMT3) and building a model based on the remaining four proteins (Proteome4 Model) resulted in a further increased concordance between the predicted slopes in both cohorts.

Genotype information has previously, as expected, been shown to harbor important information about future outcome in ADPKD. However, in our study, the full validation with all complete cohorts and benchmarking towards existing models was primarily performed in comparison to the MIC Model since genotype data was not available for all patients. Nonetheless, the additional analyses performed in the subset with genotype information confirmed both the value of genetic information in outcome prediction in ADPKD and the added benefit of the protein markers on top of combined genotype, imaging and clinical information.

This study has several limitations that need to be acknowledged. Firstly, the limited availability of genotype data restricted the depth of our models integrating this information. Secondly, creatinine measurements were obtained from a real-life setting without standardized time points, which may have introduced variability in the data. Besides, for the EC only plasma samples were available, while proteomics data was obtained from serum samples in the SC and ITC. Importantly given the observational design of our study, we cannot directly infer causality for the protein changes observed. The alterations in these proteins may either contribute to eGFR decline or result from declining kidney function. However, the analysis on eGFR correlation of protein levels in two different CKD cohorts provides at least a first insight into this question. While the fact that the majority of proteins does not reach a significant correlation with eGFR in the IgAN cohort may be a consequence of limited cohort size, it is important to note, that many of these proteins are indeed far from the significance threshold and partly even show opposite directions of correlation compared to ADPKD (e.g. APCS, SERPINA6, SERPINF1 or PCOLCE).

Proteomics analyses of biofluids hold a high potential to improve patient stratification. However, this potential has not been fully exploited to date. Serum proteome-derived biomarkers are still constrained, despite their promise to augment those obtained from urine proteomics, especially in ADPKD. Our research advances the area by discovering serum proteome markers to predict outcome in ADPKD and contributes to paving the way towards simply accessible blood-bourne parameters and developing prediction models that consolidate proteomic and clinical data to anticipate eGFR reduction. Utilizing merely six proteomic markers, while neglecting any clinical information, nearly 30% of the variance in eGFR slope was explained−a notable accomplishment relative to prior models integrating e.g. both clinical data and TKV. Research that utilized more extensive predictor sets. Furthermore, our models were robustly compared against existing outcome prediction underlining the independent predictive information contained in the proteome panel. Besides, the models were confirmed using two different cohorts: the ITC from the German AD(H)PKD registry and the EC from the Dutch DIPAK cohort, thereby strengthening the robustness of our findings. Nonetheless, despite these encouraging results, more research towards clinical implementation optimal outcome prediction in ADPKD is needed. All resulting models are still far from explaining 100% of the variance in

eGFR slope. In this regard, integration of genotype data for all patients and additional biomarkers which are currently under evaluation such as copeptin or MCP1 as well as innovative approaches including metabolomics with our proteome models will be of high interest. Addressing the question how therapeutic interventions such as tolvaptan influence the proteome patterns is another important point which will be addressed in follow-up studies based on continued sample collection in our cohorts. Besides, it will now be crucial to turn the panel of the six proteins into targeted assays with absolute quantification, e.g. based on ELISA or targeted mass spectrometry. Such targeted panels will then be the crucial basis towards multicenter prospective validation. This strategy based on the findings of the study at-hand will finally result in proteome-based markers, which beyond mere description enter routine clinical care, a central contribution to therapeutic decision-making and patient counseling in ADPKD.

## Methods

This study adheres to the International Conference on Harmonization Good Clinical Practice guidelines, and both clinical characterization and biosampling were approved by the Institutional Review Board of the University of Cologne (NCT02497521, DRKS00008910). Written informed consent was obtained from all participants.

### Study Design and Participants ADPKD cohort

The Screening Cohort (SC) and the Internal/Temporal cohort (ITC) were based on the German AD(H)PKD registry. This registry enrolls patients with an established diagnosis (based on imaging criteria or genetic diagnostics) of ADPKD in CKD stages 1–4 since 2015. Annual clinical and laboratory parameters were collected while imaging parameters were collected as applicable as well as genetic parameters during the first visit. Total Kidney Volume (TKV) at baseline is assessed using standardized magnetic resonance imaging (MRI). Patients were advised to present in a fasted state.

Only MIC 1 patients were included in the cohorts underlying the analyses of this study. The SC contained 257 baseline serum samples from 257 patients, the ITC contained 620 serum samples from 462 patients. SC and ITC share 169 patients, for which SC contains the baseline samples and ITC contains follow-up samples of the same patients.

Further validation of the models developed in the SC was achieved by making use of the DIPAK Cohort as the External Cohort (EC)[39]. This comprehensive validation approach provided a way to evaluate the robustness and generalizability of our predictive models. DIPAK Cohort was collected using EDTA-plasma, which differs from the serum samples collected in the SC and ITC. The DIPAK Cohort analyzed in this study consisted of 219 baseline samples from 219 patients.

Samples that could be affected by an intervention (e.g. tolvaptan usage, dialysis, nephrectomy and kidney transplant) were removed from further analyses (see eGFR slope calculations below and Fig. S1 for further details).

### Characteristics IgA nephropathy cohort

Serum samples obtained from patients with immunoglobulin A nephropathy (IgAN) who were included into the randomized STOP-IgAN trial were used as controls. Samples were collected at the beginning of the 6-month run-in phase. STOP-IgAN study results have been published previously (ClinicalTrials.gov number NCT00554502, Rauen et al., NEJM 2015). STOP-IgAN was approved by the leading ethics committee at the RWTH Aachen University Hospital (# EK159/07) and local ethics committee at each of the participating centers.

### Data analyses

All the analyses were performed within the statistical environment R (version 4.3.2). Figures were also generated in R with the ggplot2 (version 3.4.4), ggpubr (version 0.6.0) and ComplexHeatmap (version 2.18.0) packages.

### eGFR slope calculations

Following eGFR calculation by the CKD-EPI equation[40], yearly eGFR decrease or eGFR slope per patient were calculated by using a robust linear model (eGFR $= \alpha + \beta \times$ date). This provided a daily eGFR decrease which was multiplied by 365.25 to reach an annual eGFR loss. To ensure that clinical interventions, such as tolvaptan use, dialysis, kidney transplant, or nephrectomy would not influence our eGFR slope calculation, eGFR values affected by these confounders were removed from our modeling approach. In the German AD(H)PKD registry (Screening and Internal/Temporal Cohorts), 1758 eGFR measurements from 252 patients were removed due to interventions. More specifically, 1713 measurements from 241 patients were removed due to Tolvaptan usage. The rest of the removals were due to dialysis ($n_{measurement} = 4$ from $n_{patients} = 3$), nephrectomy ($n_{measurement} = 30$ from $n_{patients} = 6$), kidney transplants ($n_{measurement} = 11$ from $n_{patients} = 2$) and the minimum threshold of 3 eGFR measurements per patient ($n_{measurement} = 601$ from $n_{patients} = 477$).

In the DIPAK study (External Cohort), 258 eGFR measurements from 55 patients were removed due to Tolvaptan administration. While dialysis, nephrectomy, and kidney transplant led to no removal, the minimum threshold of 3 eGFR measurements resulted in the removal of 29 eGFR measurements from 19 patients.

Remaining 5371 eGFR measurements from 578 patients in the German AD(H)PKD registry (SC and ITC) and 1194 eGFR measurements from 180 patients in the In DIPAK study (EC) were used for calculating the eGFR slope. Finally, eGFR slopes that were greater than 5 or below $-10$ mL/min/1.73m$^2$ per year were removed from further analyses.

### Proteomic Data

Screening and validation proteomic data were processed separately in the CECAD proteomic facility, and therefore proteomic data were subjected to preprocessing (normalization, imputation, and batch effect correction) separately as explained below. After preprocessing, common gene names corresponding protein IDs in SC and ITC were selected for further analyses performed on SC. Samples were prepared following a modified SP3 protocol[41] performed on a Chronect Robotic RSI automated liquid handling system (AxelSemrau) for Screening Cohort (SC), an Integra Assist plus (Integra) for Internal/Temporal Cohort (ITC), and a Miocrolab Star M (Hamilton) for External Cohort (EC), foregoing the peptide cleanup on the second day. Instead, beads were removed after acidification and samples either cleaned by using mixed-mode StageTips for SC[42], or directly loaded onto Evotips following the recommended vendor protocol for ITC and EC.

Samples were analyzed on either an EASY 1200 nLC coupled to a Q Exactive HFx (both Thermo Scientific) for SC or an Evosep One (Evosep) coupled to an Orbitrap Exploris 480 with FAIMS pro (both Thermo Scientific) for ITC and EC.

### Screening proteome (screening cohort) and IgAN proteome

Samples were injected onto an in-house packed 40 cm pulled tip column (75 μm inner diameter, filled with 2.7 μm Poroshell EC120 C18, Agilent). Separation took place on a 30 min gradient running 0.1 % formic acid (eluent A) against 80 % acetonitrile, 0.1 % formic acid (eluent B). Gradient started at 8 % B and increased to 35 % over 30 min followed by washing and equilibration to standard conditions, all with a constant flow of 250 nl/min. The mass spectrometer was operated in data-independent acquisition covering the mass range between 350 and 1200 m/z with 15 variable windows. Each cycle, a single MS1 scan with 45k resolution was followed by MS2 scans at 30k resolution using stepped NCE of 25.5, 27, and 30. For library generation, a pool was generated from all samples and high pH fractionated on an 1 h gradient using an Infinity 1260 LC (Agilent). Resulting 48 fractions were concatenated into a total of 24 samples and analyzed on the identical

setup used for sample analysis. The library was build using Spectronaut 14.7 using standard settings for library building in directDIA using the Human Uniprot reference proteome including isoforms (downloaded 04.01.2021). It contained 9643 precursors from 2346 proteins. Finally, samples were searched against the library using DIA-NN 1.7.12[43] using the same FASTA file. Standard settings were used with the additional "−report-lib-info" command line input.

Screening Proteome: Protein Groups (PG) were quantified with the Diann package (version 1.0.1). There were 776 PGs identified in the raw data. Filtering $q$-value (≤0.01) followed by removal of PGs containing more than 80% NAs resulted in the data containing 398 PGs. These steps were followed by data normalization (justvsn from vsn package, version 3.70.0) and imputation (sampling from the 5th percentile of the data). PCA was performed on the processed proteome data. Fig. S10A revealed outlier samples ($n = 7$) that were separated from the main cluster. After removal of outliers, batch effect correction (ComBat function from sva package, version 3.50.0) was performed (Fig. S10B).

IgAN Proteome: Protein Groups (PG) were quantified with the Diann package (version 1.0.1). There were 776 PGs identified in the raw data. Filtering $q$-value (≤0.01) followed by removal of PGs containing more than 80% NAs resulted in the data containing 383 PGs. These steps were followed by data normalization (justvsn from vsn package, version 3.70.0) and imputation (sampling from the 5th percentile of the data). PCA was performed on the processed proteome data. Outlier samples ($n = 1$) were detected with PCA (not shown) and removed.

### Validation proteome (internal/temporal and external cohorts)

Samples were injected using the 60 SPD chromatography method and the FAIMS set to −50 V compensation voltage with inner and outer electrode temperature kept constant at 99.5 °C and 85 °C, respectively. The mass spectrometer resolution of both MS1 and MS2 were set to 15k resolution and was running in data independent acquisition mode with 30 % normalized collision energy. The mass range from 400 to 880 m/z was covered in 30 staggered windows of 16 m/z each, resulting in effectively 8 m/z windows after deconvolution using ProteoWizard[44]. For library generation, a pool was generated from all samples and high pH fractionated on an 1 h gradient using an Infinity 1260 LC (Agilent). Resulting 48 fractions were concatenated into 12 samples and analyzed on the identical setup used for sample analysis but running the mass spectrometer in DDA. MS1 resolution was set to 60k, MS2 resolution to 15k with 20 s using 30 % normalized collision energy and 1.4 Th isolation width. The library was afterwards build using Fragpipe 16.0 and its predefined library building workflow with standard parameters and the Human Uniprot reference proteome including isoforms (downloaded 04.01.2021). Afterwards, the resulting library was combined with two other HpH serum libraries to increase analytical depth. The resulting library contained 18461 precursors from 1727 proteins. Finally, samples were searched against the library using DIA-NN 1.8.1[43] with reannotation activated using the same FASTA as for library building and the additional command line "−report-lib-info".

Filtering $q$-value (≤0.01) resulted in identified 731 PGs. Then, the removal of PGs containing more than 80% NAs resulted in the final data containing 338 PGs. These steps were followed by data normalization (justvsn from vsn package, version 3.70.0) and imputation (sampling from the 5th percentile of the data). PCA was performed on the processed proteome data. Fig. S11A revealed outlier samples ($n = 5$) that were separated from the main cluster. After removal of outliers, batch effect correction was performed (Fig. S11B) (ComBat function from sva package, version 3.50.0).

### Characteristics of screening proteome

The mass distribution of the proteins corresponding to 398 PGs identified in Screening Proteome were compared with the mass distribution of all known human proteins in UniprotKB (as of February 2024). Masses were calculated by mw function from Peptides package (version 2.4.6) and density plots comparing these distributions were generated. All detected proteins and the remaining 257 samples were used to generate a PCA and a heatmap, in which patients were sorted according to their eGFR and PGs were clustered distance to 1-correlation (Pearson) and method to average.

### Enrichment analyses

All functional enrichment analyses were performed with gconvert (to convert protein IDs to ENSG IDs) and gost (to obtain functional enrichment) functions from gprofiler2 package (version 0.2.2). To identify enriched parent GO:BP terms, calculateSimMatrix (to calculate similarity score of enriched terms) and reduceSimMatrix (to calculate semantic similarity score) function from rrvgo package (version 1.14.1) were utilized.

### Enrichment of screening proteome

To evaluate the functions represented by our approach, the detected 398 PGs were subjected to an enrichment analysis, with false discovery rate (FDR) adjusted $p$-value of 0.001. Then, by calculating semantic similarity scores, the clustering of similar terms and simplified visualization of the functional space of our proteome were accomplished.

### Enrichment of selected features and patients

First, PGs identified by the LIMMA and wLASSO procedure (LIMMA and LASSO set) were subjected to functional enrichment with FDR adjusted $p$-value of 0.1 and background was assigned to all detected PGs.

Second, a heatmap was generated. Both proteins and samples were clustered by assigning distance to 1-correlation (Pearson) and method to average. Protein clusters were subjected to the functional enrichment as described above. Patient clusters and their characteristics were compared by using two different tests. While continuous variables (such as age, eGFR and slope) were compared among the clusters by $t$-test with Bonferroni adjustment ($t$-test function from rstatix package version 0.7.2), categorical variables (such as sex) were compared to the whole cohort by proportions test (prop.test function from stats package version 4.3.2).

### Identifying univariate and multivariate associations to eGFR slope

To identify proteins, which associate with eGFR slope within our cohort, we considered two distinct approaches: Linear Models for Microarray Data (LIMMA package, version 3.58.1) and weighted Least Absolute Shrinkage and Selection Operator (weighted LASSO, glmnet package, version 4.1–8). The LIMMA set was defined by proteins that associate with eGFR slope with an FDR < 0.05. Weighted LASSO (wLASSO) was used to identify multivariate sets of features that associate with eGFR slope. Here 100 training and test datasets were generated and within each the lambda optimized using a 10-fold CV. Features identified in at least 75% of the models were selected. By using Pearson correlation (rcorr function from Hmisc package, version 5.1-1), eGFR dependency of those selected features are tested on SC and IgAN Cohort.

### Development of predictive models for eGFR decline

To develop and validate predictive models for future kidney function decline we utilized the features selected by the LASSO set (Proteome Model). To test for robustness 100 training/test (2:1 ratio) datasets were generated and the R² of the test data recorded.

A second linear regression model based on clinical parameters such as age, sex, eGFR, and Mayo imaging class was generated to compare its accuracy directly to the Proteome Model. Finally, a third model combining the two previous models was established.

Two additional models were generated which extended both the Clinical and Combined Model using genotype information. To directly

compare models, sample numbers were adjusted where necessary. We generated another model (Proteome4 Model) by removing 2 proteins from the LASSO set which have a high percentage of missing values in ITC and EC.

Lastly, we implemented the Mayo Imaging Classification (MIC) Model[7], and compared its predictive capability to our models by comparing (i) the numeric slope predictions (RMSE function from caret package, version 6.0-94), (ii) the stratified slope predictions (rapid and stable progressors, roc function from pROC package, version 1.18.5) and (iii) future eGFR predictions.

### Reporting summary
Further information on research design is available in the Nature Portfolio Reporting Summary linked to this article.

## Data availability
The data underlying individual figures are provided in the source data file in an anonymized fashion. The full proteomics and clinical data supporting the findings of this study are not publicly archived to ensure data protection of study participants and minimize the risk of re-identification. Access to the dataset may be granted upon direct request to the corresponding authors depending on the nature of research questions aligning with the aims of the study to which participants provided consent and the ability to ensure data protection. To ensure this goal, data sharing will depend on a signed bilateral data transfer agreement. Interested researchers should contact the corresponding authors at roman-ulrich.mueller@uk-koeln.de or philipp.antczak@uk-koeln.de, including a description of the intended use and any institutional affiliations. The corresponding author will respond within 10 business days of receiving a complete request. Access will be provided only to individuals or institutions that meet the conditions outlined above. The data will remain available for a minimum of ten years following publication, unless otherwise restricted by law or institutional policy. Source data are provided with this paper.

## Code availability
The code is available at https://github.com/handeaydogan/slope-models-adpkd.

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

## Acknowledgements

We thank Cornelia Böhme and Serena Greco-Torres for excellent support with study coordination and sample handling. We are grateful to have received samples from the Groningen site of the DIPAK cohort for external validation. Clinical database generation was supported by "clinicalsurveys.net" (Sebastian Heimann, Jörg Janne Vehreschild). We thank the CECAD Proteomics Facility for sample preparation and mass spectrometric analyses. RUM was supported by the Ministry of Science North Rhine-Westphalia (Nachwuchsgruppen.NRW 2015-2021), the German Research Foundation (DFG DI 1501/9-2, DFG MU 3629/6-1, FOR 5547/1), the Marga and Walter Boll Foundation, and the PKD Foundation. SA was supported by the Köln Fortune Program (Faculty of Medicine, University of Cologne), CECAD-Rotationsprogramm and KFH-Stiftung. PA and RUM received support from the joergbernards-Stiftung as well as Köln Fortune and CECAD (funded by the Deutsche Forschungsgemeinschaft DFG under Germany's Excellence Strategy - EXC 2030 - 390661388). The Dept. II of Internal Medicine received research funding from Otsuka Pharmaceuticals and Thermo Fisher Scientific. We acknowledge support for the Article Processing Charge from the DFG (German Research Foundation, 491454339). This work was supported via DFG large invest grants (DFG Großgeräteantrag INST 1856/71-1 FUGG & INST 216/1070-1 FUGG). We would like to thank all patients participating in this study.

## Author contributions

H.B. and S.A. contributed equally to this work. S.A. acquired funding, curated data, prepared biological samples. H.B. and S.A. curated the data. H.B. performed the analysis. J.W.L. performed masspectrometic measurements and provided proteomics methodology. R.U.M. and P.A. conceptualized and supervised the study and acquired funding. R.U.M. contributed to data curation. R.U.M., F.G., S.A. and T.R. organized the respective clinical cohorts and provided clinical data and biosamples. H.B., S.A., P.A., and R.U.M. wrote the original draft of the manuscript. All authors reviewed and approved the final version.

## Funding

## Competing interests

All authors declare no competing interests.
