## [Transparent Peer Review file · Nature Communications]

Developing serum proteomics based prediction models of disease progression in ADPKD

Corresponding Author: Professor Roman-Ulrich Muller

Version 0:

Reviewer comments:

Reviewer #1

(Remarks to the Author)

The need of prediction models for CKD progression in ADPKD is with no doubts of high clinic relevance.

The variability of kidney outcome among ADPKD individuals from the same family and the poor reliability of current prediction models made this study of high interest.

The present study aims to apply a serum proteomics approach to detect a dataset of molecules useful to predict kidney disease progression in ADPKD. A screening cohort and further validations cohorts of ADPKD patients have been used in this study.

My concerns:

Authors "aimed to examine the potential of serum proteomics for improved risk stratification in ADPKD", as indicated in the abstract. However, data interpretation, as stated in the abstract, is not supported by results: authors cannot exclude that changes in serum proteome of ADPKD patients are the effect of eGFR decline per sé, in fact the study does not include a control group of patients with variable eGFR due to other causes.

Methods:

- As state above, why the study does not include a control group consisting of individuals with eGFR decline due to other causes should be defined.
- eGFR slope calculation: Authors estimated annual eGFR slope. Please define the mean/median period of observation for the definition of mean annula eGFR decline.

Results:

- Identification of proteins associated with annuale GFR slope: the authors indicate 4 serum proteins releted to eGFR decline. In lines 187-88, authors comment as following. "..... highlighting their potential significance in the pathophysiology of 187 renal function deterioration". The study is not designed to show the casual relationship between serum changes and eGFR decline: the changes may represent the effect of eGFR decline.

- Moreover, did the author correlate serum proteome to changes of renal volume? Besides eGFR decline, the increase in kidney volume is another parameter used in ADPKD clinical studies to address disease progression.

Discussion:

- Authors should highlight that proteomics is a valid approach for a broad screening of biological fluids; to date, the application of proteomics to urine and plasma samples of ADPKD patients have provided limited applications in the clinical setting.
- Authors show a set of circulating proteins that may predict CKD progression in ADPKD. Please provide a revision of the literature on the possible association of these molecules and CKD progression in other chronic kidney disease models: this analysis can help the readers to understand whether the association you find in this study is ADPKD-specific.

Reviewer #2

(Remarks to the Author)

Aydogan Balaban and colleagues explore the application of serum proteomics for risk stratification of disease severity in patients with ADPKD. They analyzed the serum proteome of 257 patients using a semi-automated mass spectrometer, validating their findings on an internal cohort of 466 patients and an external cohort of 221 patients. The proteomic model was compared to observed eGFR slopes and the Mayo Imaging Classification, a commonly used imaging-based stratification model.

The authors identified 29 proteins associated with eGFR slopes, with six proteins optimizing the protein-based linear prediction model. Notably, proteins related to immune response, lipoprotein metabolism, metabolic processes, and transport were linked to changes in eGFR slope. The study utilized the AD(H)PKD registry in Germany, with DIPAK serving as the external cohort. The cohorts appear representative of the ADPKD population.

Major Critiques:

1. Model Validation: The adjusted R^2 is reported at 0.31. It would be beneficial for the authors to compare this with other models beyond the Mayo Imaging Classification to contextualize whether an R^2 of 0.31 is considered robust.
2. Patient Stratification: Considering that younger patients may not exhibit an eGFR slope >3 despite being at risk of rapid progression, it is advisable to stratify patients based on observed GFR rates (fast vs. slow) and predicted risks using both the Mayo Imaging Classification and PROPKD scores. It would be insightful to align proteomics biomarkers thresholds with these known progression rates.
3. Clinical Application: A critical clinical question is the prediction of ESRD onset. The authors have predicted future eGFR but this information is relegated to a supplemental figure. Clarity on the accuracy and error of this prediction is needed. Additionally, Figure S7 requires better explanation both within the text and its legend.
4. Proteomic Indicators: Is the mere presence of specific proteins indicative of a worse outcome? Establishing thresholds for these proteins, and correlating them with known biological activities in ADPKD, would enhance the utility of the findings. Furthermore, is it possible for each proteome to be predictive on its own?
5. Impact of Therapy: In patients who received disease-modifying therapy, such as tolvaptan, were there any changes in the proteomics profile? This would be crucial for understanding the impact of such treatments on disease progression.

Minor Points:

1. Cohort Consistency: There is a discrepancy in the number of patients in the external cohort (221 vs. 219 mentioned), with results from only 173 patients being used. Clarification is needed.
2. Patient Numbers: The abstract mentions 257 patients, yet 264 are referred to in the results, with only 214 analyzed post-QC and outlier removal. It would be beneficial to specify the number of patients included in the model directly in the abstract.
3. Exclusion Criteria: The methods section should clarify whether patients who received tolvaptan, underwent dialysis, or other significant interventions were excluded from the analysis post these events.
4. Follow-up Data: It would enhance clarity to include median follow-up time in the results, in addition to the median number of eGFR measurements and the median time between these measurements.
5. Data Presentation: In the tables, presenting some results to one decimal point, such as percentage of females, age, and eGFR, would improve readability.

Version 1:

Reviewer comments:

Reviewer #1

(Remarks to the Author)

The revised manuscript is much improved.

I appreciate the following changes:

- The inclusion of patients with kidney disease from another cause;
- the detection of proteins correlating with the eGFR in both models (cases and controls) and with ADPKD only;
- The incorporation of table S1;
- The comments on the potentialities of proteomics studies in ADPKD and the reasons why previous studies have provided limited implementations to date;
- Comments, along the discussion section, on the potential meanings of detected proteins, based on previous studies;
- comments on the limitations of the studies.

I have only a minor comments:

I would incorporate in the abstract the name of the 6 proteins predicting ADPKD progression.

Reviewer #2

(Remarks to the Author)

well done. no additional comments from my end.

We have carefully addressed all the concerns identified by the reviewers in the point-by-point response below, and we believe that the revision process has indeed improved the quality of the manuscript. While this manuscript was under review, the KDIGO 2025 Clinical Practice Guideline for the Evaluation, Management, and Treatment of Autosomal Dominant Polycystic Kidney Disease (ADPKD) was published online in January 2025¹. This guideline states in regard of the topic of our study: “Given the costs and complexities of kidney imaging and genotyping in ADPKD, considerable efforts have been made to develop better prognostic and treatment-monitoring urine and serum biomarkers in people with ADPKD. However, at this stage, most such biomarkers have not outperformed the traditional means of monitoring kidney function—SCr, and cystatin C levels—and the development of improved biomarkers is an area for future research.” Accordingly, in the research recommendations, the guideline lists: “Identify and validate better urine and serum biomarkers of ADPKD progression.” And further the need to “Develop a model that includes multiple imaging, and genetic, clinical, and biomarker inputs to better predict disease outcomes in ADPKD.” We feel that this strongly underlines the importance of our data for the field and do now quote the guideline in the revised version of the

Some figures and tables were prepared and provided for the reviewers only at this points and are thus provided as reviewer only (RO) in the point-by-point reply. Provided line and page numbers are for the clean version of the manuscript.

Reviewer #1 (Remarks to the Author):

The need of prediction models for CKD progression in ADPKD is with no doubts of high clinic relevance. The variability of kidney outcome among ADPKD individuals from the same family and the poor reliability of current prediction models made this study of high interest. The present study aims to apply a serum proteomics approach to detect a dataset of molecules useful to predict kidney disease progression in ADPKD. A screening cohort and further validations cohorts of ADPKD patients have been used in this study.

My concerns:

Authors “aimed to examine the potential of serum proteomics for improved risk stratification in ADPKD”, as indicated in the abstract. However, data interpretation, as stated in the abstract, is not supported by results: authors cannot exclude that changes in serum proteome of ADPKD patients are the effect of eGFR decline per sé, in fact the study does not include a control group of patients with variable eGFR due to other causes.

We agree that CKD control cohorts are of much use in molecular analyses. This is explicitly the case when focussing on pathophysiology, e.g. proteomics to identify pathways altered in a specific disease. This cannot be fully done within a single cohort suffering from one specific kidney disease and comparison to healthy controls does not address the question of eGFR-associated changes. Consequently, we have,

following the reviewer's suggestion, added measurements from a separate cohort as described below. However, the question of pathophysiology-associated changes in the proteome was not the primary focus of our study. We primarily focussed on outcome prediction (eGFR slope) and apologize if this did not get clear enough in the manuscript. For this question, it is not central whether a marker is also changed by eGFR itself in a potentially disease-specific manner. However, it is indeed crucial that a potential impact of eGFR on the markers (through collinearity) is addressed considering the possibility that the markers only reflect eGFR and do not have added predictive value (and would thus be meaningless). This point is addressed in our study (equally to or more stringently compared to many other studies in the field) by including eGFR in the multivariate models. Doing this fully accounts for this risk and shows in our case that the proteome-based markers are indeed independent from eGFR in their predictive capacity (while their levels may indeed be eGFR dependent, either in a disease-specific or general manner). This is the central aspect of the study and is backed up by its very stringent design including two validation cohorts. Nonetheless indeed, we also point to the potential of serum proteome data to shed light towards pathophysiology, e.g. regarding the function of the proteins in the predictive panel or the whole list of proteins associated with eGFR slope. We agree that especially these analyses benefit from knowing if an individual marker is eGFR dependent or not. To answer this question we have added two types of analyses. Firstly, we analyzed eGFR dependency within the ADPKD cohort (Table S9). Secondly, to address the reviewer's request we have added data from an additional CKD cohort suffering from IgA nephropathy to allow for an insight into which proteins are altered in an eGFR dependent manner and for which this may be ADPKD-specific (Figure S4, Table S10). The results are listed in the results section (page 8, lines 208-220) and further discussed in the discussion section (page 15, lines 372-378). Besides, we have improved the discussion section including more insight into published data on the markers identified in our study (page 15-16, lines 378-386).

Additionally, to indicate the specific strengths of our study and compare the methodologies, we have performed an extensive literature search for biomarker studies in ADPKD (MEDLINE search March 01, 2025, Table RO1, below) identifying 16 studies. 3 of these used omics-technologies, the remaining 13 tested targeted individual markers. Only 2 studies included external validation cohorts. As to control groups 4 studies included healthy probands and 3 studies CKD cohorts while the remaining 10 studies included no control cohorts. 11 of the 16 published studies indeed predicted future eGFR decline, while 5 studies only performed cross sectional analyses (i.e. essentially can only provide data on eGFR dependency but not outcome prediction). 10 of the 11 studies predicting eGFR decline do not contain data on a CKD control cohort (a point which was now added to our manuscript based on the reviewer's suggestion). Follow-up available varies from 1 to 11 years, underlining the meaningful nature of our mean follow-up times from 5.9 to 8 years in the individual cohorts (see Table RO1). Cohort sizes vary from less than 100 to 1000 patients, placing our study among the largest cohorts examined to date. As an example the only study reaching

a considerably longer median follow-up of 11.2 years included only 79 patients and does not use a validation cohort². Indeed, when looking at studies that predict outcome (i.e. have access to eGFR slope) and include validation cohorts our study is – in our knowledge - the largest dataset to date. This highlights the added benefit of our study now including a separate CKD cohort and makes it quite unique through the combination of 2 validation cohorts, a CKD control cohort and availability of long-term follow-up data for eGFR. We did not add any of these considerations (nor Table RO1) to the manuscript since we felt this would have led to a rather lengthy discussion, but are open, of course, to add individual aspects if the reviewers/editors would consider this helpful.

Table R01: List of biomarker studies in ADPKD.

Study (Author, Year)	Patients (ADPKD)	Predictive for eGFR decline (including median FU time (years) if applicable)?	Validation Cohort	Control Groups	Study Type	Biomarker	Source / Link
Meijer et al., 2011	102	No (cross-sectional only)	No	-	Clinical (cross-sectional)	Copeptin	https://pubmed.ncbi.nlm.nih.gov/20930090/
Boertien et al., 2012	79	Yes (11.2)	No	-	Clinical (prospective)	Copeptin	https://pubmed.ncbi.nlm.nih.gov/22523115/
Boertien et al., 2013	241	Yes (8.5)	No	-	Clinical (prospective)	Copeptin	https://pubmed.ncbi.nlm.nih.gov/23089511/
Casteleijn et al., 2015	55	Yes (max. 2.8,)	No	-	Clinical (prospective; Copeptin/Osmol.)	Copeptin, Osmolality	https://pubmed.ncbi.nlm.nih.gov/25926129/
Kistler et al., 2013	41 + 251 (Validation)	No	Yes (external validation)	Healthy + CKD	Clinical (proteomics study; CE-MS in urine)	Urinary Peptidome	https://pubmed.ncbi.nlm.nih.gov/23326375/
Pejchinovski et al., 2017	241 (CRISP)	Yes (mean 9.9)	No (internal CV)	CKD	Clinical (prospective; proteomics in urine)	Urinary Peptidome	https://pubmed.ncbi.nlm.nih.gov/27382111/
Messchendorp et al., 2017	104	Yes (mean 3.82)	No	-	Clinical (prospective; multiple urine markers)	β2-Microglobulin, MCP-1	https://pubmed.ncbi.nlm.nih.gov/29725632/
Messchendorp et al., 2019	152 + 104 (Training + Validation)	Yes (mean 2.43)	Yes (second cohort)	-	Clinical (prospective; biomarkers + external validation)	β2-Microglobulin, MCP-1	https://pubmed.ncbi.nlm.nih.gov/31600749/
Segarra-Medrano et al., 2020	130	Yes (median not specified, at least 10)	No	CKD	Clinical (prospective; urine markers + CKD controls)	KIM-1, VEGF, MCP-1	https://pubmed.ncbi.nlm.nih.gov/32905289/
Heida et al., 2021	583	Yes (4)	No (single cohort)	-	Clinical (prospective; metabolic: U/P urea ratio)	U/P Urea Ratio	https://pubmed.ncbi.nlm.nih.gov/33504546/
Houske et al., 2023	48	No (candidate markers identified)	No	Healthy	Clinical (cross-sectional; metabolomic profile)	Metabolomic Profile	https://pubmed.ncbi.nlm.nih.gov/37141147/
Arjune/Oehm et al., 2023	389	Yes (median not specified, max 3)	No	Healthy	Clinical (prospective)	Copeptin	https://pubmed.ncbi.nlm.nih.gov/37915893/
Arjune/Späth et al.	184	Yes (median not specified, max 4.5)	No	Healthy	Clinical (prospective)	DKK3	https://pubmed.ncbi.nlm.nih.gov/38186869/
Lacquantiti et al., 2013	52	Yes (n.e., max 2)	no	no	Clinical (prospective)	Apelin, Copeptin	https://pubmed.ncbi.nlm.nih.gov/23973863/
Chonchol et al., 2017	1002	No	No	-	Clinical (cross-sectional)	FGF-23	https://pubmed.ncbi.nlm.nih.gov/28705885/
Hayek et al., 2019	649	Yes (n.e. max 3)	No	-	Clinical (prospective)	SuPAR	https://pubmed.ncbi.nlm.nih.gov/31171572/

Methods:

- As state above, why the study does not include a control group consisting of individuals with eGFR decline due to other causes should be defined.

Please see our response to the general point above. This criticism was fully addressed.

- eGFR slope calculation: Authors estimated annual eGFR slope. Please define the mean/median period of observation for the definition of mean annual eGFR decline.

This is indeed an important piece of information which was previously presented as a sentence in the first paragraph of the results section but contained an error in the sentence. This was now corrected (page 6, line 167-168) and – to add additional useful information – we included the following table as Table S1.

Table S1 : Overview of eGFR value availability and timeframe.

	Screening Cohort (SC)	Internal/Temporal Cohort (ITC)	External Cohort (EC)
Median Follow-up time (days [years])	2489 [6.8]	2329 [6.4]	2205 [6]
Mean Follow-up time (days [years])	2933.8 [8]	2783.3 [7.6]	2166.8 [5.9]
Median number of eGFR measurements	10	9	7
Median time between measurements (days [years])	183 [0.5]	163 [0.4]	365 [1]

Results:

- Identification of proteins associated with annuale GFR slope: the authors indicate 4 serum proteins related to eGFR decline. In lines 187-88, authors comment as following. "..... highlighting their potential significance in the pathophysiology of 187 renal function deterioration'. The study is not designed to show the casual relationship between serum changes and eGFR decline: the changes may represent the effect of eGFR decline.

We acknowledge that our study is observational in nature and does not establish causality between the identified serum proteins and eGFR decline. The associations observed may reflect either a contributory role of these proteins in renal function deterioration or be a consequence of declining kidney function. To provide an insight to the reader we have now added the analyses on eGFR correlation of the 29 proteins identified (including all the proteins used in the models) as Table S9 and S10. Besides, we amended the previous wording by:

- 1) Deleting the following wording from the results section: “highlighting their potential significance in the pathophysiology of renal function deterioration”
- 2) Adding the following sentence to the limitations section: “Importantly given the observational design of our study, we cannot directly infer causality for the protein changes observed. The alterations in these proteins may either contribute to eGFR decline or result from declining kidney function. However, the analysis on eGFR correlation of protein levels in two different CKD cohorts provides at least a first insight into this question.” (page 18, lines 448-452)

- Moreover, did the author correlate serum proteome to changes of renal volume? Besides eGFR decline, the increase in kidney volume is another parameter used in ADPKD clinical studies to address disease progression.

We appreciate the reviewer's suggestion to examine the correlation between the serum proteome and changes in renal volume. However, serial MRI-based total kidney volume (TKV) measurements are not part of standard clinical care for ADPKD and are no longer recommended for assessing disease severity, as outlined in the ERA Statement on Tolvaptan in ADPKD³. Consequently, longitudinal TKV data are not available for our cohort. Cross Sectional correlation with TKV, which would in principle be possible in our cohorts, could be added, but we feel that the additional insight obtained is limited and by far outcompeted by the prediction of eGFR decline.

Discussion:

- Authors should highlight that proteomics is a valid approach for a broad screening of biological fluids; to date, the application of proteomics to urine and plasma samples of ADPKD patients have provided limited applications in the clinical setting.

We appreciate the reviewer's comment and agree that proteomics is a valuable approach for broad screening of biological fluids, offering insights into disease-associated biomarkers. While proteomics has been applied to urine and plasma samples from ADPKD patients, its clinical translation remains limited due to variability in findings, lack of standardization, and the need for further validation in larger cohorts. Our study contributes to addressing these challenges by employing stringent modeling approaches, utilizing independent validation cohorts, and integrating eGFR slope as a robust measure of disease progression. These methodological strengths enhance the reliability and potential applicability of our findings. However, we acknowledge that further work is needed to bridge the gap between discovery and clinical implementation. Future research should focus on targeted absolute quantification of key biomarkers and prospective longitudinal studies to establish their predictive value and standardize their use in clinical practice. To ensure clarity, we revised the Discussion section (page 14-15, lines 346-348) to explicitly highlight both the potential and current limitations of proteomics in ADPKD and emphasize the necessary next steps for clinical translation. This thought is also addressed by clearly mentioning that – while our study solves some of the questions by including a thorough modeling and validation approach – clinical implementation will require prospective studies using targeted panels of the biomarkers identified.

- Authors show a set of circulating proteins that may predict CKD progression in ADPKD. Please provide a revision of the literature on the possible association of these molecules and CKD progression in other chronic kidney disease models: this analysis

can help the readers to understand whether the association you find in this study is ADPKD-specific.

We appreciate the reviewer's suggestion to examine whether the circulating proteins identified in our study are specifically associated with ADPKD progression or if they reflect broader CKD mechanisms based on the existing literature. We had originally included more information on these points in the manuscript but had to exclude this due to limitations in word count. Below, you will find our full assessment (both in a written form and in Table RO2). We have added the key thoughts based on this assessment to the updated discussion section (page 15-16, lines 378-386), but would, of course, be open to add more if the reviewers/editors think this would help and word count limits allow.

GPX3 and Afamin (AFM), both involved in oxidative stress regulation, have been linked to kidney function decline across various CKD etiologies^{4,5}. In ADPKD, increased oxidative stress has been proposed as a contributor to cyst growth and disease progression, making these antioxidant-associated proteins particularly relevant. The predictive value of GPX3 and Afamin in our study suggests mechanisms related to oxidative stress may play a pronounced role in ADPKD. GPX3 has not previously been indicated as a biomarker of eGFR slope, but has been shown to be positively correlated with eGFR in CKD before (similar to our data in ADPKD). The situation is more complex for AFM. Here, previous work implicated AFM as a predictor of eGFR slope (similarly to our study), but the correlation with eGFR was opposite to our data indicating disease-specific findings.

PEDF (SERPINF1) and FHR-1 (CFHR-1), involved in vascular and immune regulation, are known to be altered in CKD⁶. PEDF has been associated with vascular remodeling in diabetic nephropathy⁷, while CFHR-1 is widely studied in IgA nephropathy due to its role in complement activation⁸. Both proteins were previously implicated to be positively correlated with eGFR, in line with our findings in ADPKD, but have never been examined regarding eGFR slope. Our findings suggest that these proteins might also be involved in ADPKD-specific processes, including cyst-associated angiogenic changes and complement-mediated inflammation.

Chemerin (RARRES2) and Kindlin-3 (FERMT3), which influence inflammatory and immune cell adhesion processes, have been broadly associated with systemic inflammation and metabolic disturbances in CKD. In ADPKD, their altered levels may modulate disease progression, e.g. based on metabolic shifts within cystic kidneys or immune dysregulation in cyst-associated inflammation. Interestingly, while Chemerin's relation to eGFR itself is in line with previously published work, the association with eGFR slope shows the opposite direction in patients with ADPKD. We did not find any literature on Kindlin-3 in the context of CKD or ADPKD.

While these proteins have been discussed in broader CKD pathophysiology, especially data as to their relation to eGFR slope is very limited and only available for

two of the markers. Besides, the cohorts examined were often very limited in size and used different methodologies as to marker identification. Consequently, this review of the literature provides first indications towards ADPKD-specific and more general findings but will need further validation in future studies.

Table RO2: Review of the literature on biomarkers identified in this study. Table shows whether the markers have previously been associated with eGFR slope and eGFR in the published literature, in CKD in general or in ADPKD specifically, as well as the direction of correlation.

Genes	Proteins	Slope (↓)		eGFR (↓)		Organism	Links
		Our Study	Literature	Our Study	Literature		
GPX3	GPX3	↓	x	↓	↓ (CKD, plasma)	mouse	https://pubmed.ncbi.nlm.nih.gov/29244159/
						human	https://pubmed.ncbi.nlm.nih.gov/15980946/
						human	https://pubmed.ncbi.nlm.nih.gov/10050081/
						human	https://www.nature.com/articles/pr19962385
SERPINF1	PEDF	↑	x	↑	↑ (CKD, serum)	human	https://pmc.ncbi.nlm.nih.gov/articles/PMC9559055/
					↑ (DKD, serum)	human	https://pmc.ncbi.nlm.nih.gov/articles/PMC4223434/
CFHR1	FHR-1	↑	x	↑	↑ (IgAN & ADPKD, plasma)	human	https://pubmed.ncbi.nlm.nih.gov/28637589/
						human	https://pmc.ncbi.nlm.nih.gov/articles/PMC5611987/
RARRES2	Chemerin	↑	x	↑	↑ (CKD, serum)	human	https://pubmed.ncbi.nlm.nih.gov/25945605/
			↓ (CKD, plasma)		x	human	https://pmc.ncbi.nlm.nih.gov/articles/PMC10116780/
AFM	Afamin	↓	x	↓	↑ (CKD, serum)	human	https://pubmed.ncbi.nlm.nih.gov/37835901/
			↓ (CKD, plasma)		x	human	https://pmc.ncbi.nlm.nih.gov/articles/PMC10116780/
FERMT3	Kindlin-3	↑	x	↓	x	x	x

Reviewer #2

Aydogan Balaban and colleagues explore the application of serum proteomics for risk stratification of disease severity in patients with ADPKD. They analyzed the serum proteome of 257 patients using a semi-automated mass spectrometer, validating their findings on an internal cohort of 466 patients and an external cohort of 221 patients. The proteomic model was compared to observed eGFR slopes and the Mayo Imaging Classification, a commonly used imaging-based stratification model. The authors identified 29 proteins associated with eGFR slopes, with six proteins optimizing the protein-based linear prediction model. Notably, proteins related to immune response, lipoprotein metabolism, metabolic processes, and transport were linked to changes in eGFR slope. The study utilized the AD(H)PKD registry in Germany, with DIPAK serving as the external cohort. The cohorts appear representative of the ADPKD population.

Major Critiques:

1. Model Validation: The adjusted R^2 is reported at 0.31. It would be beneficial for the authors to compare this with other models beyond the Mayo Imaging Classification to contextualize whether an R^2 of 0.31 is considered robust.

We appreciate the comment regarding the adjusted R^2 value of 0.31 in our model and the suggestion to contextualize its robustness by comparing it with other predictive models beyond the Mayo Imaging Classification. The adjusted R^2 value indicates the proportion of variability in the dependent variable that is explained by the independent variables in the model, adjusted for the number of predictors. An adjusted R^2 of 0.31 suggests that approximately 31% of the variability in eGFR slope is accounted for by our model. In clinical predictive modeling, especially within heterogeneous populations like those with ADPKD, such an R^2 value is considered moderate and reflects the complex, multifactorial nature of disease progression. While specific R^2 values for the Mayo Imaging Classification are not commonly reported, studies have demonstrated its predictive utility. For instance, a recent study involving 618 patients showed that the Mayo could generally predict kidney failure and eGFR decline rates, though there was notable interindividual variability within each class⁹. However, one key challenge that comes with Mayo, in addition to the need of imaging data, is the fact that Mayo Class 2 cannot be distinguished based on volume alone and requires expert review in every case. Importantly, also other key studies in the field have previously shown that models containing TKV indeed show, while being the gold standard in the field, only a limited R^2 comparable or actually lower to our findings in this regard^{10,11}. This aspect was added to the discussion of the manuscript. Furthermore the PROPKD Score effectively stratifies patients into low-, intermediate-, and high-risk categories for progression to end-stage renal disease. While the original study introducing the PROPKD score did not report an R^2 value, it highlighted the score's capacity to predict renal outcomes, with median ages for end-stage renal disease onset varying significantly across risk groups. Nonetheless, the Mayo Imaging Classification is

currently considered the best model available and was thus chosen as our direct comparator. This is again underlined by the recent KDIGO ADPKD guidelines¹ defining it as gold-standard of care (Recommendation 1.4.2.1). The PROPKD score can be employed when imaging data are not available or if imaging data is not sufficient and further evidence towards rapid progression is required². In line with this, we show that adding genotype data to the models – despite the limited availability in our cohorts – further improves model performance and should thus be considered in the future. In conclusion, we feel we have indeed compared to the best available model (Mayo Imaging Class) and have pointed this out in the discussion section (page 16-17, lines 411-412 & 414-420).

2. Patient Stratification: Considering that younger patients may not exhibit an eGFR slope >3 despite being at risk of rapid progression, it is advisable to stratify patients based on observed GFR rates (fast vs. slow) and predicted risks using both the Mayo Imaging Classification and PROPKD scores. It would be insightful to align proteomics biomarkers thresholds with these known progression rates.

We appreciate this point and added more analyses using this cutoff comparing the MIC and Combined Model (but refrained from doing so for PROPKD considering the limited availability of genotype data). Figure S9 contains the ROC curves for predicting an eGFR slope > 3 ml/min/1.73m². AUCs are 0.82 for the Combined Model and 0.69 for the MIC model, respectively (page 14, lines 337-340).

As to thresholds, these cannot be defined based on untargeted proteomics considering its nature as a method of relative quantification. This is also why we point out very clearly in the outlook section that validation using targeted panels in a future prospective study is urgently required (page 18-19, lines 480-485).

3. Clinical Application: A critical clinical question is the prediction of ESRD onset. The authors have predicted future eGFR but this information is relegated to a supplemental figure. Clarity on the accuracy and error of this prediction is needed. Additionally, Figure S7 requires better explanation both within the text and its legend.

We appreciate the reviewer's request to clarify the accuracy and error of our eGFR predictions. However, this analysis is not only shown in the supplements but introduced in Figure 2D. To further contextualize these findings, we included an additional accuracy metric of root mean squared error (RMSE) to quantitatively compare the predictive performance of the models (page 12, lines 285-286 and Figure legend 2D). As to the figure, the y-axis represents the difference between observed and predicted eGFR values ($\Delta eGFR = \text{Observed} - \text{Predicted}$), where values closer to zero indicate higher prediction accuracy. Each data point corresponds to an individual patient, with models color-coded as follows: Proteome Model (blue), Combined Model (dark blue), and Mayo Imaging Classification (MIC) Model (orange). The fitted regression lines and confidence intervals illustrate model performance, with larger deviations from zero representing greater prediction error.

Figure S7 (now Figure S8) uses the same approach to depict the predictive performance of our models by adding the results of this analysis obtained in the Internal/Temporal Cohort (ITC) and the External Cohort (EC). Our results indicate that the proteome containing models show lower prediction errors, while the MIC Model demonstrates greater variability in the SC and ITC. This advantage of the proteome-containing models is not yet reached in the EC which was based on EDTA-plasma instead of serum as discussed in the manuscript.

To enhance clarity, we have revised the legend of Figure 2D (page 11-12, lines 270-279) and Figure S8 to explicitly define key terms and explain their clinical significance. Additionally, we provide a clearer interpretation of the results, including the implications of prediction errors and how different models perform in identifying patients at risk of rapid eGFR decline.

4. Proteomic Indicators: Is the mere presence of specific proteins indicative of a worse outcome? Establishing thresholds for these proteins, and correlating them with known biological activities in ADPKD, would enhance the utility of the findings. Furthermore, is it possible for each proteome to be predictive on its own?

We appreciate the reviewer's comment regarding whether the mere presence of specific proteins is indicative of a worse outcome and whether individual proteomes can independently predict patient outcomes. Mass spectrometry-based proteomics provides relative quantification rather than absolute concentrations, meaning that proteins falling below the detection limit are not necessarily absent but simply undetectable within the sensitivity of the assay. Therefore, establishing strict thresholds for protein presence is not feasible, and the mere presence or absence of a protein does not necessarily indicate a worse outcome. Instead, we focus on relative abundance and its correlation with disease progression. However, to gain an insight whether the presence of specific proteins in our quantification correlates with worse outcomes, we examined missing values in the screening cohort and assessed their association with eGFR slope:

- FERMT3 – 19.4% missing, no significant correlation with eGFR slope.
- RARRES2 – 22.2% missing, significant negative correlation with eGFR slope.
- Other proteins (SERPINF1, GPX3, AFM and CFHR1) – 0% missing, detected in all patients.

This suggests that for RARRES2 (Chemerin), detection correlates with disease progression (patients with detectable levels tend to have a more negative eGFR slope), but this does not imply that the mere presence of the protein drives disease progression. Instead, it reinforces a biological association between RARRES2 levels and ADPKD severity. We have added this finding to the results section (page 10, lines 241-243).

While individual proteins may correlate with disease progression, we argue that no single protein alone is sufficient to robustly predict individual patient outcomes. Instead, our model integrates multiple proteomic markers, leveraging a multi-biomarker approach to enhance predictive performance. Given the complexity of ADPKD pathophysiology, a single-protein-based approach would likely lack the sensitivity and specificity required for individualized prognostication. To further enhance the clinical utility of these findings, we acknowledge that absolute quantification of these proteins will be a crucial next step. Future targeted mass spectrometry assays (e.g., parallel reaction monitoring, PRM) will enable precise concentration measurements, allowing for more defined clinical cutoffs and improved risk stratification.

As to biological activities we had originally included more thoughts but had to remove these due to word limits. We feel that the discussion section on biological functions addresses this question, but would, of course, be open to add more there if the reviewers /editors feel this would be helpful and word limits allow.

We understand that the reviewer is wondering whether a single individual proteome would predict an individual's progression. Once the models are developed, as we have done here, a single proteome can be simply used within our model and provide a single predicted slope value for that individual. The error across individual proteome prediction is indicated by our analyses in Figure 2D and S8. Due to technical differences within Mass Spectrometry approaches, there might be a higher variance when using different spectrometers or sample processing pipelines. This is why we also suggest that the selection of proteins be tested in an absolute quantification.

5. Impact of Therapy: In patients who received disease-modifying therapy, such as tolvaptan, were there any changes in the proteomics profile? This would be crucial for understanding the impact of such treatments on disease progression.

This is indeed an interesting question. However, our cohort was not designed to specifically obtain data before and on tolvaptan yet. Nonetheless, we checked for how many of the patients we had such data resulting in a subcohort of 20 patients. There were no proteins differentially regulated in this cohort for the comparison of last time point before tolvaptan and the first time point on tolvaptan (see Figure RO1 below). We feel that this is probably also a result of limited size of this subcohort as well as the decrease in eGFR over time as a confounder in this longitudinal analysis and have thus not added the data to the manuscript. Answering this question is, however, a very important point for the follow-up of this study (we have added this thought to the outlook section (page 18, lines 477-480) and sample collection is ongoing to address this point in the future.

Figure R01: Characteristics of ADPKD Patient from ITC, before and after tolvaptan usage. A) Clinical characteristics of ADPKD patients. **B)** Volcano plot of the detected proteins ($n=338$) compared between before and after tolvaptan usage of each patient ($n=21$). Each point represents detected proteins. No protein was differentially regulated in a statistically significant manner. **C)** Heatmap of identified proteins which are associated with eGFR slope ($n=28$, in columns) and samples ($n=42$, in rows), and corresponding clinical parameters. The samples were sorted according to tolvaptan usage and eGFR.

Minor Points:

1. *Cohort Consistency: There is a discrepancy in the number of patients in the external cohort (221 vs. 219 mentioned), with results from only 173 patients being used. Clarification is needed.*

We appreciate the reviewer's attention to the cohort consistency and apologize that, while the numbers were correct, these differences did not get fully clear from our previous explanations. In our study, we initially had 221 proteomic samples from the external cohort. After outlier removal, two samples were excluded, leaving 219 samples. Further filtering was applied to ensure robust slope calculations. We excluded patients with prior tolvaptan use and included at least three eGFR measurements to reliably determine the eGFR slope. After applying these criteria, 173 samples remained for the final analysis. This step was necessary to maintain data quality and consistency in assessing proteomic associations with eGFR decline. We have revised the manuscript (page 6, lines 164-167) to clarify this workflow and ensure transparency in how the final sample size was determined.

2. *Patient Numbers: The abstract mentions 257 patients, yet 264 are referred to in the results, with only 214 analyzed post-QC and outlier removal. It would be beneficial to specify the number of patients included in the model directly in the abstract.*

The differences arise because different models include different numbers of patients based on availability and quality control criteria. To ensure consistency, we reported the number of patients included in the proteome model in the abstract's methods part

(page 2, lines 38-40), as it represents the primary focus of our study. In the results section, we provide a more detailed breakdown, including patients analyzed post-quality control (QC) and outlier removal across different models (page 6, lines 160-166)

3. Exclusion Criteria: *The methods section should clarify whether patients who received tolvaptan, underwent dialysis, or other significant interventions were excluded from the analysis post these events.*

Clarification regarding the exclusion of patients who received tolvaptan, dialysis, kidney transplants, or other significant interventions in our analysis is an important point. This is explicitly addressed in our manuscript (Page 4, eGFR Slope Calculation Section) & Supplementary Methods (Suppl. Page 2, eGFR Slope Calculation Section) and we have tried to point this out more clearly in the revised manuscript:

“Samples that could be affected by an intervention (e.g. tolvaptan usage, dialysis, nephrectomy and kidney transplant) were removed from further analyses (see eGFR slope calculations below).” (page 4, 108-110)

“To ensure that clinical interventions, such as tolvaptan use, dialysis, kidney transplant, or nephrectomy would not influence our eGFR slope calculation, eGFR values affected by these confounders were removed from our modeling approach.” (page 4, 122-124)

4. Follow-up Data: *It would enhance clarity to include median follow-up time in the results, in addition to the median number of eGFR measurements and the median time between these measurements.*

We thank the reviewer for this suggestion and have included the following table contained in the supplementary file as Table S1.

Table S1 : Overview of eGFR value availability and timeframe

	Screening Cohort (SC)	Internal/Temporal Cohort (ITC)	External Cohort (EC)
Median Follow-up time (days [years])	2489 [6.8]	2329 [6.4]	2205 [6]
Mean Follow-up time (days [years])	2933.8 [8]	2783.3 [7.6]	2166.8 [5.9]
Median number of eGFR measurements	10	9	7
Median time between measurements (days [years])	183 [0.5]	163 [0.4]	365 [1]

5. Data Presentation: In the tables, presenting some results to one decimal point, such as percentage of females, age, and eGFR, would improve readability.

We have revised the tables to present percentages (e.g., female distribution), age, and eGFR values to one decimal place in Tables 1, S2 and S8. This adjustment will enhance clarity while maintaining accuracy in data presentation.

References

- 1 KDIGO 2025 clinical practice guideline for the evaluation, management, and treatment of autosomal dominant polycystic kidney disease (ADPKD): executive summary Torres, Vicente E. et al. , *Kidney International*, Volume 107, Issue 2, 234 - 254
- 2 Wendy E. Boertien, Esther Meijer, Debbie Zitteema, Marjan A. van Dijk, Ton J. Rabelink, Martijn H. Breuning, Joachim Struck, Stephan J.L. Bakker, Dorien J.M. Peters, Paul E. de Jong, Ron T. Gansevoort, Copeptin, a surrogate marker for vasopressin, is associated with kidney function decline in subjects with autosomal dominant polycystic kidney disease, *Nephrology Dialysis Transplantation*, Volume 27, Issue 11, November 2012, Pages 4131–4137, <https://doi.org/10.1093/ndt/gfs070>
- 3 Müller RU, Messchendorp AL, Birn H, Capasso G, Cornec-Le Gall E, Devuyst O, van Eerde A, Guirchoun P, Harris T, Hoorn EJ, Knoers NVAM, Korst U, Mekahli D, Le Meur Y, Nijenhuis T, Ong ACM, Sayer JA, Schaefer F, Servais A, Tesar V, Torra R, Walsh SB, Gansevoort RT. An update on the use of tolvaptan for autosomal dominant polycystic kidney disease: consensus statement on behalf of the ERA Working Group on Inherited Kidney Disorders, the European Rare Kidney Disease Reference Network and Polycystic Kidney Disease International. *Nephrol Dial Transplant*. 2022 Apr 25;37(5):825-839. doi: 10.1093/ndt/gfab312. PMID: 35134221; PMCID: PMC9035348.
- 4 Pang P, Abbott M, Abdi M, Fucci QA, Chauhan N, Mistri M, Proctor B, Chin M, Wang B, Yin W, Lu TS, Halim A, Lim K, Handy DE, Loscalzo J, Siedlecki AM. Pre-clinical model of severe glutathione peroxidase-3 deficiency and chronic kidney disease results in coronary artery thrombosis and depressed left ventricular function. *Nephrol Dial Transplant*. 2018 Jun 1;33(6):923-934. doi: 10.1093/ndt/gfx304. PMID: 29244159; PMCID: PMC5982720.
- 5 Kaur, R.; Krishan, P.; Kumari, P.; Singh, T.; Singh, V.; Singh, R.; Ahmad, S.F. Clinical Significance of Adropin and Afamin in Evaluating Renal Function and Cardiovascular Health in the Presence of CKD-MBD Biomarkers in Chronic Kidney Disease. *Diagnostics* 2023, 13, 3158. <https://doi.org/10.3390/diagnostics13193158>
- 6 Hui E, Yeung CY, Lee PC, Woo YC, Fong CH, Chow WS, Xu A, Lam KS. Elevated circulating pigment epithelium-derived factor predicts the progression of diabetic nephropathy in patients with type 2 diabetes. *J Clin Endocrinol Metab*. 2014 Nov;99(11):E2169-77. doi: 10.1210/jc.2014-2235. Epub 2014 Aug 28. PMID: 25166721; PMCID: PMC4223434.

7 Hui E, Yeung CY, Lee PC, Woo YC, Fong CH, Chow WS, Xu A, Lam KS. Elevated circulating pigment epithelium-derived factor predicts the progression of diabetic nephropathy in patients with type 2 diabetes. *J Clin Endocrinol Metab.* 2014 Nov;99(11):E2169-77. doi: 10.1210/jc.2014-2235. Epub 2014 Aug 28. PMID: 25166721; PMCID: PMC4223434.

8 Medjeral-Thomas NR, Lomax-Browne HJ, Beckwith H, Willicombe M, McLean AG, Brookes P, Pusey CD, Falchi M, Cook HT, Pickering MC. Circulating complement factor H-related proteins 1 and 5 correlate with disease activity in IgA nephropathy. *Kidney Int.* 2017 Oct;92(4):942-952. doi: 10.1016/j.kint.2017.03.043. Epub 2017 Jun 30. PMID: 28673452; PMCID: PMC5611987.

9Bais T, Geertsema P, Knol MGE, van Gastel MDA, de Haas RJ, Meijer E, Gansevoort RT; DIPAK Consortium. Validation of the Mayo Imaging Classification System for Predicting Kidney Outcomes in ADPKD. *Clin J Am Soc Nephrol.* 2024 May 1;19(5):591-601. doi: 10.2215/CJN.000000000000427. Epub 2024 Feb 26. PMID: 38407866; PMCID: PMC11108249.

10 Urinary Biomarkers to Identify Autosomal Dominant Polycystic Kidney Disease Patients With a High Likelihood of Disease Progression Messchendorp, A. Lianne et al. *Kidney International Reports*, Volume 3, Issue 2, 291 - 301

11 Sita Arjune, Simon Oehm, Polina Todorova, Ron T Gansevoort, Stephan J L Bakker, Florian Erger, Thomas Benzing, Volker Burst, Franziska Grundmann, Philipp Antczak, Roman-Ulrich Müller, Copeptin in autosomal dominant polycystic kidney disease: real-world experiences from a large prospective cohort study, *Clinical Kidney Journal*, Volume 16, Issue 11, November 2023, Pages 2194–2204,